# An Overview of the Design of Chitosan-Based Fiber Composite Materials

Chen Xue and Lee D. Wilson *

Department of Chemistry, University of Saskatchewan, 110 Science Place, Saskatoon, SK S7N 5C9, Canada; chx257@mail.usask.ca
* Correspondence: lee.wilson@usask.ca; Tel.: +1-306-966-2961

**Abstract:** Chitosan composite fibrous materials continue to generate significant interest for wastewater treatment, food packaging, and biomedical applications. This relates to the relatively high surface area and porosity of such fibrous chitosan materials that synergize with their unique physicochemical properties. Various methods are involved in the preparation of chitosan composite fibrous materials, which include the modification of the biopolymer that serve to alter the solubility of chitosan, along with post-treatment of the composite materials to improve the water stability or to achieve tailored functional properties. Two promising methods to produce such composite fibrous materials involve freeze-drying and electrospinning. Future developments of such composite fibrous materials demands an understanding of the various modes of preparation and methods of structural characterization of such materials. This review contributes to an understanding of the *structure–property* relationships of composite fibrous materials that contain chitosan, along with an overview of recent advancements concerning their preparation.

**Keywords:** chitosan; biopolymer; composite fibers; electrospinning; structure–property relationships

## 1. Introduction

### 1.1. Chitosan

Chitosan (chi), β-(1,4)-aminoglucopyranose contains randomly distributed N-acetyl-glucosamine and glucosamine residues (cf. Figure 1). Chi is a copolymer made from deacetylation of chitin, which is the second most abundant biopolymer in nature next to cellulose, and is mainly found in invertebrates, fungi, and yeasts [1]. Chitosan can be made from chitin by enzymatic or chemical processes [2]. The enzymatic process represents a controlled, non-degradable process to hydrolyze N-acetamide bonds in chitin utilizing chitin deacetylases, leading to the well-defined chitosan such as chitosan oligomers [2]. The chemical process is a favorable method to produce chitosan for commercial usage due to the low cost and scalable mass production [2]. Chitosan is a non-toxic, biodegradable, biocompatible material with many interesting chemical and physical properties, along with attractive biological functionalities, such as antimicrobial, antioxidant, antitumor activities, etc. [2] The degree of acetylation (DA) of chitosan is lower than 40% and its solubility in acidic media, making chitosan easier to manipulate using solution-based methods than cellulose [3]. Moreover, chitosan behaves as a cationic polyelectrolyte with a dissociation constant ($pK_a$) value of ~6.3 because of the presence of the free amine groups within its biopolymer structure that can undergo protonation in acidic media [4]. The occurrence of protonation of the glucosamine groups of chitosan facilitate the potential to form polyelectrolyte complexes (PECs). Chitosan also been reported to dissolve in several organic solvents and solvent mixtures such as trifluoroacetic acid (TFA), 1,1,1,3,3,3-hexafluoroisopropanol (HFIP), chloroform, glycine chloride, aqueous acetic acid/dimethylformamide, aqueous acetic acid/DMSO, lithium hydroxide/urea, HFIP/formic acid, and THF/DMF [5]. However, the solubility of chitosan in these solvents or solvent mixtures is limited and strongly

depends on the DA and molar mass of chitosan. In addition, the free amine groups on C-2 (cf. Figure 1) affords the option to chemically modify chitosan by various synthetic strategies, which will be discussed herein. Chitosan has a good water swelling ability, which is beneficial for wound dressing. These advantages of chitosan over other biopolymers make chitosan a versatile precursor for materials design and development.

R= H or COCH$_3$

**Chitosan**

**Cellulose**

**Figure 1.** Chemical structures of chitosan and cellulose.

Chemical and physical properties of chitosan, such as crystallinity, solubility (in acidic medium), and reactivity, depend on several structural features of the chitosan biopolymer. These features include the average molecular weight, degree of deacetylation of the chitosan materials, as well as the local and global distribution of N-acetyl-glucosamine units along the chain [1]. For example, the degree of acetylation (DA) of chitosan and the local and global distribution of N-acetyl-glucosamine units along the chitosan chain contribute to inter-/intra-molecular interactions that involve hydrogen bonding and van der Waals forces. In turn, the bonding arrangement determines the morphology of chitosan and the crystalline nature of the biopolymer network [6]. Furthermore, the crystallinity of chitosan relates to the accessibility of the hydroxyl and amine groups of chitosan, which determines the chemical reactivity of chitosan, and the hydration properties of the biopolymer in aqueous media, including its ability to form complexes with cationic species [7]. The solubility of chitosan in acidic media can be evaluated from its *pKa* value, which can be calculated using Katchalsky's equation, as given in Equation (1): [4,8]

$$pKa = pH + \log\left(\frac{1-\alpha}{\alpha}\right) = pKo - \frac{e\Delta\psi(\alpha)}{k_B T} \tag{1}$$

$\Delta\psi(\alpha)$ is the difference for the electrostatic potential between the surface of chitosan and the reference state at a distance from the axis in the rod-like model, where $\alpha$ is the degree of dissociation, $k_B$ is the Boltzmann constant, T is the absolute temperature, and *e* is the electron charge. The intrinsic dissociation constant of the ionisable groups, *pKo*, can be obtained by extrapolating the *pKa* value to $\alpha = 1$, where the polymer becomes uncharged (or completely deionized). This value (*pKo*) is ~6.5, which is independent of the degree of dissociation; however, the *pKa* value is highly dependent on the degree of dissociation [4]. The degree of dissociation relates to the accessibility of amine groups of chitosan, which in turn depends on the DA level of chitosan and the distribution of N-acetyl-glucosamine units along the chitosan chain [6,8]. Chitosan in acidic media displays a semi-rigid behavior depicted by a persistence length, which is marginally determined by the DA of the sample [2]. The source of chitin used as the precursor and the method employed for the deacetylation of chitin in the chemical process can influence the physicochemical properties of chitosan [2]. The chitin precursor used can be α-, β-, or

γ-chitin. The occurrence of variable unit cell parameters clearly differentiate α- and β-, but γ-chitin is considered to be a mixture or a distortion of the α- and β-forms [7,9]. According to Hajji et al. [10], chitosan prepared from β-chitin has a lower DA level than α-chitin but a higher degree of degradation because β-chitin has higher reactivity than α-chitin. Chitosan prepared from α-chitin demonstrated a slightly higher crystallinity than that prepared from β-chitin [10]. Methods of deacetylation of chitin in the chemical process are either homogenous or heterogeneous. For the heterogeneous method, chitin is immersed in a hot concentrated NaOH solution (with a temperature above 40 °C) for several hours, where chitosan is then obtained as an insoluble material with DA that ranges from ca. 1% to 15% [2]. For the homogeneous method, chitin is dispersed in concentrated NaOH (with a mass ratio between NaOH, $H_2O$, and chitin being 10:15:1) at 25 °C for 3 h or more, then the dispersion is cooled in crushed ice at 0 °C. The DA of the product is ca. 10% [2]. The heterogeneous method leads to an irregular distribution of N-acetyl-glucosamine units along the chitosan chain, whereas; the homogeneous method results in a uniform distribution [11]. The modality of the deacetylation may also affect the DA, molar mass, and viscosity of the biopolymer in solution [2].

### 1.2. Chitosan Composite Fibers

As described above, chitosan is a versatile platform for material design. Various forms of chitosan have been developed to meet the requirements for different applications. For example, beads, microspheres, nanoparticles, and coatings have been used for drug delivery, enzyme immobilization, gene delivery, surface modification, and textile finishes. Films have been used for dialysis membrane and antimicrobial membranes. Powders of chitosan have been used directly as adsorbents for water and various form of water contaminants (heavy metals and dyes). Chitosan solutions have been applied as coagulant-flocculants, cosmetics, bacteriostatic agents, anticoagulants, and antitumor agents. Chitosan gels have been applied for implants, coatings, and tissue engineering. Chitosan tablets and capsules have been used for disintegrating agents and delivery vehicles. Chitosan composite fibers have been widely used in food packaging, medical textiles, sutures, bone regeneration, tissue engineering, wound dressings, drug delivery, enzyme entrapment, and artificial skin [5,12]. Unlike synthetic polymers, thermal processes like melt blowing are not suitable for preparing biopolymer fibrous materials because biopolymers can be degraded at a high temperature [3]. Therefore, the non-thermal methods, such as electrospinning, solution blowing, wet spinning, and freeze-drying, are often used to produce nonwoven micro-/nano-fibrous biopolymer materials [3]. For chitosan composite fibrous materials, electrospinning and freeze-drying, are methods commonly applied in the literature.

Synthetic polymers, as additives in the preparation of chitosan composite fibrous materials to optimize chemical, structural, mechanical, morphological, and biological properties. The application of such polymer additives aide in overcoming the disadvantages (e.g., hydrophilicity, crystallinity, and brittleness) prevalent in pristine chitosan as a biopolymer scaffold [13]. The hydrophilic character of chitosan can lead to poor moisture barrier properties, which is a major drawback for certain applications in food packaging and food preservation [14]. Hydrophilic character of chitosan may lead to leaching or the loss of constituents in aqueous media, which results in poor stability of the product in solution. By contrast, the hydrophilic effects of chitosan can be beneficial for biomedical applications since it contributes to good water swelling property. [13,14] Therefore, the selection of materials for biomedical applications need to reach a relative balance between hydrophilic and hydrophobic character, which can be accomplished through chemical modification of chitosan and/or post-treatment methods (cf. Sections 2.1 and 2.4, respectively), along with synthetic polymer additives. The relative crystallinity and brittleness of chitosan can also be modified with the use of additive polymers. The crystallinity of chitosan can provide barrier properties to gases (e.g., $O_2$ and $CO_2$), which is useful for food packaging materials to extend the shelf-life of food products; while the semi-crystalline nature of chitosan leads to the brittleness in its pristine form. [13,14] A compromise can be made through the for-

mation of multilayer composites or post-treatment methods [13,14]. In the case of chitosan composite fibrous materials, their utility as active food packaging and biomedical applications, share some common properties that include biodegradability, biocompatibility, and ductility [13–15]. These properties can be fulfilled using water-soluble, biocompatible, and biodegradable synthetic polymers such as PEO [16–20], PVA [21,22], PAM [5], PLGA [23], poly($\varepsilon$-caprolactone) (PCL) [5], poly(vinyl pyrrolidone) (PVP) [24,25]. The use of such chitosan-polymer blends imparts new functionalities that include sensitivity to photooxidation [24], enhanced ion-exchange [25], conductivity [26], and other improved properties [27]. While chitosan can be easily dissolved in dilute acidic aqueous media, it has limited solubility in other solvents that limit the preparation of chitosan composite fibrous materials from polymer blends. As such, chitosan and synthetic polymers require dissolution in the same solvent. Therefore, adequate water solubility is a prerequisite condition for the selection of synthetic polymers as additives for chitosan-based materials. Miscibility of chitosan and the target synthetic polymer determines the fluid properties of the blend. In turn, the miscibility affects the component distribution, the morphology, and the physicochemical property of the corresponding product such as crystallinity [19,28]. The additive polymer can be utilized in the freeze-drying process and the electrospinning process to control the morphology, including the tuning of the physicochemical property of chitosan composite fibers. For instance, during freeze-drying, a compact lamellar structure may be formed due to strong molecular interactions among the PECs. To obtain a fibrous structure, a synthetic polymer with a flexible backbone have been used as a space-filler to attenuate the interactions among PECs by weakening the electrostatic interactions or hydrogen bonding, which leads to the formation of the fibrous structure [27,29]. During electrospinning, the benefits of using synthetic polymer additives will be discussed in the accompanying sections below. Briefly, the addition of the synthetic polymer can facilitate electrospinning by enhancing chain entanglements in the blends or by inducing molecular interactions between chitosan and the synthetic polymer.

In this review, we provide an overview of recent scientific literature that covers various modes of preparation of chitosan fiber composites that include freeze-dried and electrospun composites. This coverage provides insight on the underlying rationale relating to tuning of the physicochemical properties related to the hydrophile-lipophile balance of such composites, and control over the structure and morphology of the resulting materials. This overview highlights new approaches drawn mainly from the last five years and perspectives on the future directions of chitosan-based composite fibers.

## 2. Preparation Procedures and Methods

### 2.1. Modification of Chitosan

As mentioned previously, common methods to prepare composite micro-/nanofibrous materials can be achieved by freeze-drying or electrospinning. For both methods, the first step of the material preparation is to solubilize chitosan. Although chitosan is soluble in acidic media or partially soluble in organic solvents and organic co-solvent mixtures, as discussed in Section 1.1. An alternative approach to enhance the solubility of chitosan involves chemical modification to improve the solubility in neutral aqueous medium and also to increase the solubility of chitosan in organic solvents to adapt different technical requirements. Common synthetic approaches involve grafting chitosan with various functional groups and polymers, such as carboxymethylation, quaternization, and PEGylation [5]. Much of the modification methods utilize the hydroxyl group on C6 (sometimes with a side reaction at the hydroxyl group on C3) and the amine group on C2 of chitosan. Carboxymethylation is a common method to make chitosan soluble in aqueous media with neutral pH. Some of the approaches involve the synthesis of carboxymethylated (cm) chitosan as shown in Figure 2. It is worth noting that the difference between O- and N-carboxymethylated chitosan relate to variable inter-/intra-molecular interactions, which result in distinct reactivity and solution chemistry behaviour. This trend parallels that for other types of chemical modification. Apart from carboxymethylation, quaternization

of chitosan is a common strategy, where quaternary ammonium groups are introduced to chitosan, leading to an enhancement of the solubility of chitosan at neutral and high pH environments in aqueous media [30]. PEGylation is another important method to increase the solubility of chitosan in aqueous media [31]. In the case of chitosan with either carboxymethylation or quaternization, such ionic groups possess ionic charge at neutral and high pH in aqueous media. However, in the case of PEGylation, a non-ionic hydrophilic polymer such as poly(ethylene glycol) (PEG) was grafted onto chitosan. During PEGylation, PEG with an appropriate end group is firstly synthesized, which is then attached to chitosan. To increase chitosan solubility in organic solvents or to change the hydrophile-lipophile balance (HLB) of chitosan, the introduction of acyl, alkyl, and benzyl groups can be incorporated [5].

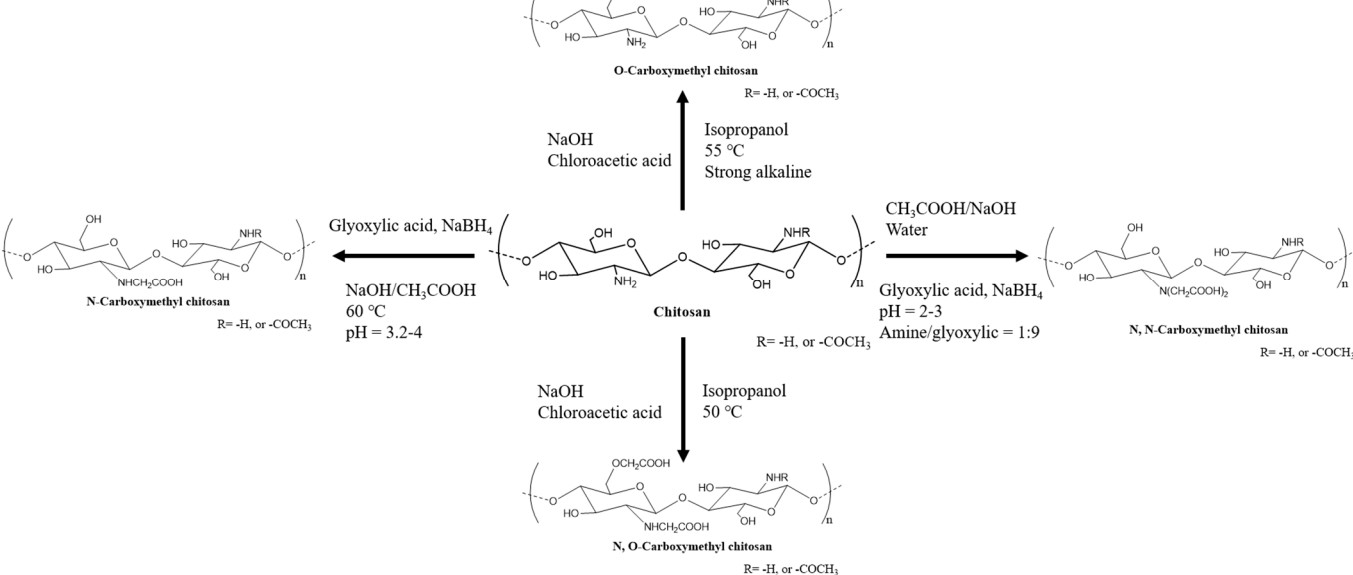

**Figure 2.** Pristine chitosan and different types of carboxymethylated chitosan [32].

During the preparation of composite fibrous materials by electrospinning, chitosan or its modified form is initially dissolved in a proper solvent. Then, the solution properties, such as polymer concentration, viscosity, surface tension, dielectric constant, conductivity, etc., are adjusted to achieve a favourable condition related to the local microstructure in solution. Thereafter, the resulting solution with a suitable local microstructure is converted into an integrated macrostructure, consisting of solid fibrous material, through electrospinning. Some examples of chitosan-based electrospun fibers are hexanoyl chitosan, PEGylated chitosan, cm-chitosan blended with other water-soluble synthetic polymers, as follows: PEO, polyacrylamide (PAM), poly(acrylic acid) (PAA), poly(vinyl alcohol) (PVA)) [21], chitosan/PVA [22], chitosan/poly(lactide-*co*-glycolide) (PLGA)/PVA [23], chitosan/PEO [16–20], chitosan/collagen [33], chitosan/agarose, chitosan/zein, chitosan/Nylon-6, etc. [21].

For fibrous materials prepared by freeze-drying, the first step is to form polyelectrolyte complexes (PECs) in the solution medium. These PECs are then assembled into solid fibrous materials by the freeze-drying method. Some examples are chitosan/acetate [34], chitosan/alginate [35], chitosan/hyaluronate [36], chitosan/poly(galacturonic acid) (PGA) [37], and so on. It is worth noting that even though various morphological forms of chitosan (e.g., films, powders, tablets, capsules, etc.) can be prepared by freeze-drying, the formation of a dispersion of PECs in the solution prior to freeze-drying is a sufficient condition to produce chitosan-based micro- or nano-fibrous materials. This relates to the mechanism of freeze-drying that will be discussed in the following section.

## 2.2. Freeze-Drying Method

### 2.2.1. Background of the Freeze-Drying Method

Freeze-drying, also known as lyophilization or cryodesiccation, is a dehydration process where water in the product is frozen and then removed by subliming the ice to vapor. There are three stages in the freeze-drying process: freezing, primary drying, and secondary drying. During the freezing stage, the sample's temperature is reduced until the initiation of the nucleation of ice crystals, which is followed by ice crystal growth [38]. This leads to the separation of most of the water (in the form of ice crystals) from a concentrated solute phase, usually containing a small amount of water [38]. During primary drying, the crystalline ice formed during freezing is removed by sublimation [38,39]. Therefore, the chamber pressure is reduced well below the vapor pressure of ice, and the shelf temperature is raised to supply the heat removed by ice sublimation [38]. At the end of the primary drying, the sample may still contain about 20% of unfrozen water in the concentrated solute phase, which is then desorbed during the secondary drying stage, usually at elevated temperature and low pressure, to finally attain the desired low moisture content of the final product [38].

The freezing stage plays an essential role in the freeze-drying method. Many studies have shown that the freezing stage greatly impacts the quality and the morphology of the final freeze-dried product [35,39–44]. Generally, freezing can be defined as the process of ice crystallization from supercooled water, since it is a type of water remaining as a liquid at the temperature below its equilibrium freezing point [39]. Supercooling is a non-equilibrium, meta-stable state, where the energy is close to the activation energy of the nucleation process. The formation of ice crystals starts from the nucleation. Ice-like clusters, resembling the structure of ice crystals, are formed by water molecules in the supercooled water through prolonged hydrogen bonds due to density fluctuations from Brownian motion [39]. The probability of forming these energetically unfavorable clusters increases as the temperature decreases [39]. Ice crystal growth is initiated when the critical mass of these clusters (nuclei) is met [39]. There are two types of nucleation, which are homogenous and heterogeneous nucleation. Heterogeneous nucleation is observed during the freeze-drying process, which suggests that ice-like clusters are formed by adsorbing layers of water on "foreign impurities" such as the surface of the container, particulate contaminants present in the water, and large molecules [45]. Ice crystal growth is controlled by the heat released from the addition of water molecules to the ice-water interface and the cooling rate which the sample undergoes exposure. Since a portion of the heat released by the ice formation (15 cal/g of the 79 cal/g of heat given off by the ice formation) can be absorbed by the supercooled water, where the rest of the heat needs to be removed by further cooling [39]. The ice morphology is based on two factors: the degree of supercooling and the freezing mechanism [39]. The degree of supercooling depends on the solution properties and process conditions. It is defined as the difference between the equilibrium ice formation temperature and the actual temperature at which ice crystal first forms [38,39]. A high degree of supercooling results in a high number of small ice crystals.

However, at a lower degree of supercooling, a lower number of large ice crystals were observed [46]. There are two basic freezing mechanisms involved in the freeze-drying process which are global supercooling and directional solidification. During the occurrence of global supercooling (like shelf-ramped freezing), the whole liquid volume has a similar level of supercooling and solidification that occurs through the already nucleated volume, which leads to spherulitic ice crystals [39]. For the directional solidification, it occurs when a small volume is supercooled, which occurs for high cooling rates that result with liquid nitrogen sample immersion. During high cooling rates, the nucleation and the development of the ice-water interface occurs almost simultaneously, where the ice-water interface moves further into non-nucleated solution leading to directional lamellar morphologies with connected pores [39,42,44,46]. As ice crystals grow in the freezing stage, the phase separation takes place between most of the water in the form of ice crystals and a concentrated solute phase. As a result, the space among ice crystals is

occupied by the concentrated solute phase [42]. If this separation does not happen, a solid solution is formed with a greatly reduced vapor pressure and the sample cannot be freeze-dried [39]. The concentrated solute phase may undergo eutectic crystallization or vitrification depending on the properties of the solutes, which result in a mixture of small crystals of ice and solute or a mixture of amorphous solutes and amorphous (unfrozen) water, respectively [39]. Other behaviors, like liquid-liquid phase separation (further phase separation of the multicomponent system in the concentrated solute phase), crystallization of amorphous solids, or amorphization from crystalline solids, may also take place, which deeply affect the physicochemical properties of the freeze-dried product. Following the completion of freezing, the concentrated solute phase adopts the morphology of the ice crystals acting as a "template". Upon removal of the "template" in the second and third stages of the freeze-drying process, the freeze-dried product with the corresponding morphology can be obtained [42,44].

### 2.2.2. Recent Development on Preparation of Freeze-Dried Chitosan Composites

Reports indicate that nonwoven fibrous materials such as sponges and foams can be prepared through freeze-drying [34–37,40,41,44,47,48]. Despite the use of variable starting materials and the types of preparation procedures among the various studies, there are two common features for the preparation of nonwoven fibrous materials. One strategy involves freezing with high cooling rates and a diluted dispersion of high molecular weight polymers as the starting material like cellulose, chitosan, alginate, poly(lactic acid) (PLA), PLGA, gelatin, etc. [34–37,40,41,44,47–49]. As discussed above, freezing with high cooling rates can result in directional solidification of the sample results in directional lamellar structure of ice crystal with connected pores. The dispersed polymers in the concentrated solute phase are rearranged into the fibrous structure or the lamellar structure according to the initial dispersion concentration [35,41]. Moreover, the use of high molecular weight polymers ensures that the microstructure created by "the ice template" remains unchanged during the second and third drying process. A schematic illustration of the formation of different morphologies of the freeze-dried high molecular weight polymer is shown in Figure 3. In the context of chitosan-based composite fibrous materials, such materials can be prepared by the freeze-drying process that follow the same mechanism. A stable diluted dispersion of chitosan-based materials can be achieved in the solution by forming PECs, as mentioned in Section 1.2. The PECs undergo freezing with high cooling rates using liquid nitrogen immersion (or the ultra-low-temperature freezer), followed by drying [34–37]. Thus, the formation of a dispersion of PECs in the solution is crucial for the preparation of chitosan-based nonwoven nano- or micro-fibrous materials.

A recent study on PECs based on the preparation of freeze-dried composite fibers by Chen et al. [50] revealed that salt concentration can significantly impact the morphology of such composite fibers. In this study, carboxymethyl cellulose (cm-cellulose)/N-2-hydroxylpropyl trimethylammonium chloride chitosan (HACC) composite fibers were prepared from a dispersion of PECs at different NaCl concentrations by freeze-drying. Morphologies and molecular dynamics simulation results for cm-cellulose/HACC fibers prepared at different concentration of NaCl are shown in Figure 4. Based on this study, a suitable salt concentration can favor the extension of polyelectrolyte polymers in solution leading to the formation of uniform fibers along with the enhancement of thermal stability.

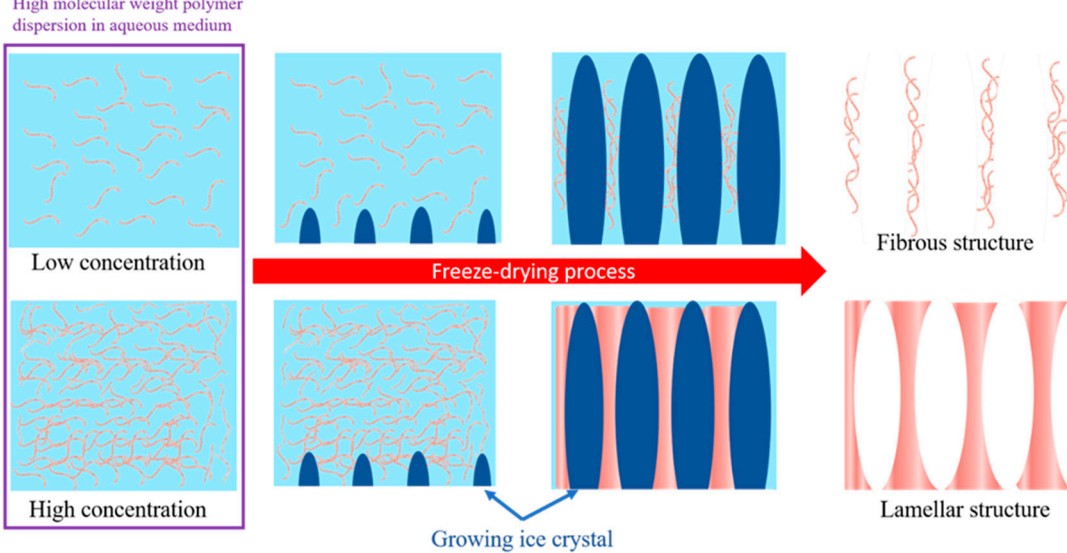

**Figure 3.** A schematic illustration of the formation of different polymer morphologies of the freeze-dried high molecular weight polymer dispersion [32].

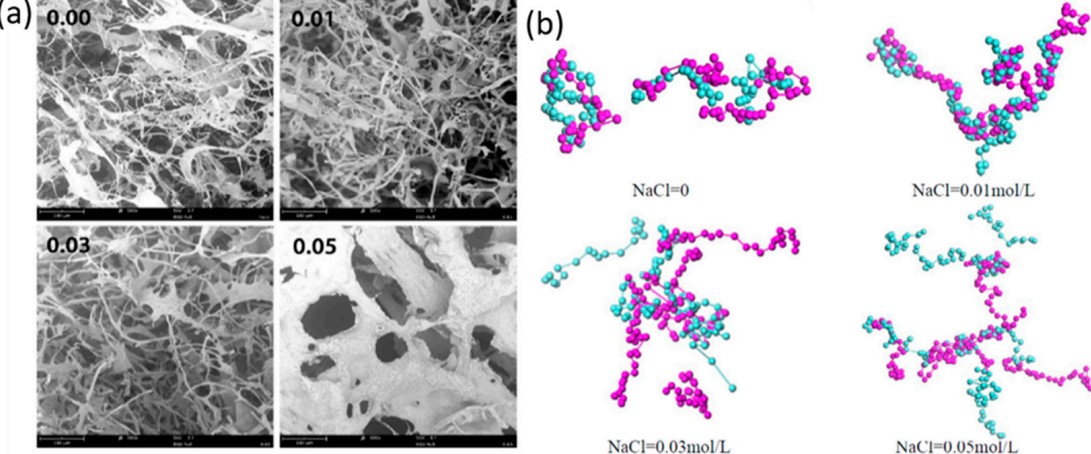

**Figure 4.** Surface morphologies (**a**) and conformation results of the molecular dynamics simulation (**b**) of cm-cellulose/HACC fibers prepared with different concentrations of NaCl. Reprinted with permission from [50].

An alternative method for preparing chitosan freeze-dried composite fibers was demonstrated by Perez-Puyana et al. [51] Chitosan/gelatin composite fibers were prepared using three different protocols in Figure 5a. The difference among the three protocols is that thermal treatment was placed at different stages. Morphologies of composite fibers prepared using different protocols are displayed in Figure 5b. It can be concluded that the composite fibers prepared by Protocol 3 (P3) is the most homogenous product, whereas the sample prepared by Protocol 2 (P2) shows significant heterogeneity. In particular, the sample prepared by P3 exhibits high porosity, large surface area/volume ratio, and great interconnectivity between pores. Moreover, different protocols can improve the mechanical properties of products.

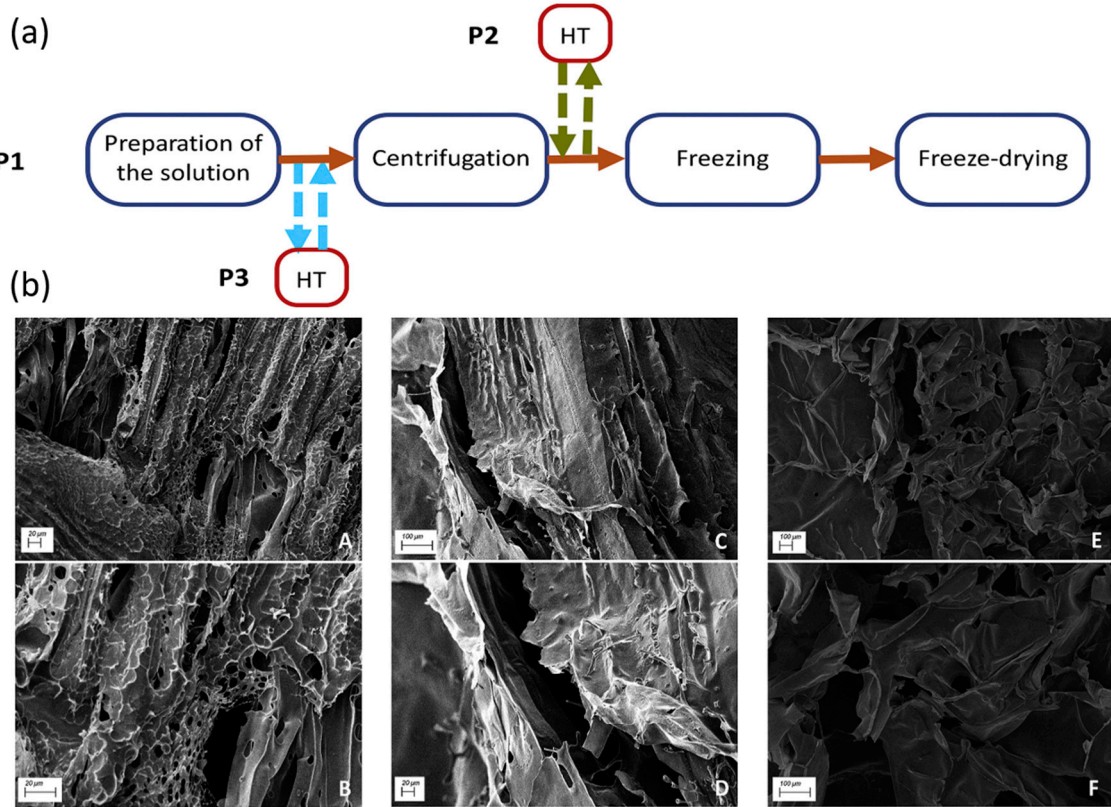

**Figure 5.** (**a**) Traditional scaffold preparation stages (Protocol 1, P1). HT means heat treatment, which is an additional step included in protocols 2 (P2) and 3 (P3) in different stages of the process (marked with dashed arrows); (**b**) SEM micrographs for the scaffolds with 50 wt.% gelatin and 50 wt.% chitosan (50GE-50CH) processed by the different protocols. (**A,B**): protocol 1. (**C,D**): protocol 2. (**E,F**): protocol 3. Reprinted with permission from [51].

### 2.3. Electrospinning Method

#### 2.3.1. Brief History of Electrospinning

Electrospinning provides a facile method to produce ultrathin fibers with diameters that extend down to the nanometer scale regime. Electrospinning may be considered a variant of the electrostatic spraying (or electrospray) process, both of which depend on using a high voltage to eject liquid jets [52]. The significant differences between electrospinning and electrospraying are the viscosity and viscoelasticity of the liquid involved that affects the behavior of the jet [53]. During electrospinning, the continuous jet results in fibers, whereas the jet breaks into droplets leading to the formation of particles during electrospraying. The earliest concept of electrospinning could be found in 1887. Charles V. Boys reported that fibers could be drawn from a viscoelastic liquid (e.g., beeswax and collodion) under an external electric field [54]. A prototype of the electrospinning setup was patented in 1902 [53]. In the late 1930s, improvements of the electrospinning setup was made by Anton Formhals aiming to commercialize electrospinning [53]. In the same period, the first example of electrospun nanofibers, "Petryanov filters", was realized in the Soviet Union to capture aerosol particles [53]. For the next 60 years, neither academia or industry has not paid much attention to electrospinning, resulting from the limitation on characterization tools to measure fibers with sub-micron diameters accurately [53]. The tide turned in the late 1990s. Several research groups, notably those led by Darrell Reneker and Gregory Rutledge [55–62], brought this "ancient technique" into the laboratory to produce nanofibers from various organic polymers as electron microscopes that can resolve nanoscale features which became available for researchers [53]. Thus, the term "electrospinning" was widely accepted and used in the literature to describe this technique. Since the beginning of the 21st century, because of its remarkable simplicity, versatility, and the

potential uses, the electrospinning method was considered as the method of choice for producing fibrous materials with nanoscale diameters [53].

### 2.3.2. Basic Theory of Electrospinning

In 1887, Charles V. Boys used an apparatus consisting of an insulated dish connected to an electrical supply [54]. Nowadays, a basic electrospinning setup consists of a high-voltage power supply, a spinneret, and a collector (a grounded conductor). It does not differ much from the first apparatus. A scheme of the basic electrospinning setup is shown in Figure 6. The high-voltage power supply can either be direct current (DC) or alternating current (AC). The spinneret, containing the polymer solution, is equipped with a blunt-tip needle, and a syringe pump that controls the polymer solution's flow rate. Many advanced electrospinning setups have been developed to regulate the alignment of electrospun nanofibers [53], to control the structure of electrospun nanofibers (coaxial setup for core-sheath and hollow nanofibers [63]), or to increase the throughput of electrospun nanofibers (multi-needle electrospinning setup [64]). The electrospinning process generally can be divided into four successive steps: (1) charging a pendant droplet to form a cone-shaped jet in an electric field; (2) elongating the charged jet; (3) stretching and thinning the charged jet under the electric field leading to the growth of bending instability (also called whipping instability); (4) solidification and collection of the jet as solid fibers on a grounded collector. Recently, comprehensive reviews regarding the theory of the electrospinning process were published by different groups [12,53]. The critical processing parameters of the electrospinning setup involve the voltage applied to the spinneret, the flow rate of the liquid, and the working distance between the tip of the spinneret and the collector. In general, the voltage applied to the spinneret determines the strength of the electric field. The flow rate of the liquid typically relates to the diameter of the fiber. Long enough working distance is required to achieve fully extended solid fibers. However, there is a complicated interplay among all the processing parameters, making the optimization of the electrospinning process difficult. Besides, the physicochemical properties of the polymer solution can also affect the electrospinning process significantly, such as the polymer molecular weight, viscosity, viscoelasticity, surface tension, dielectric constant, etc. In particular, the ability of the polymer solution to overcome the Rayleigh instability which occurs in the second step of the electrospinning process is a key factor that prevents the jet from breaking up into droplets that lead to the fiber formation.

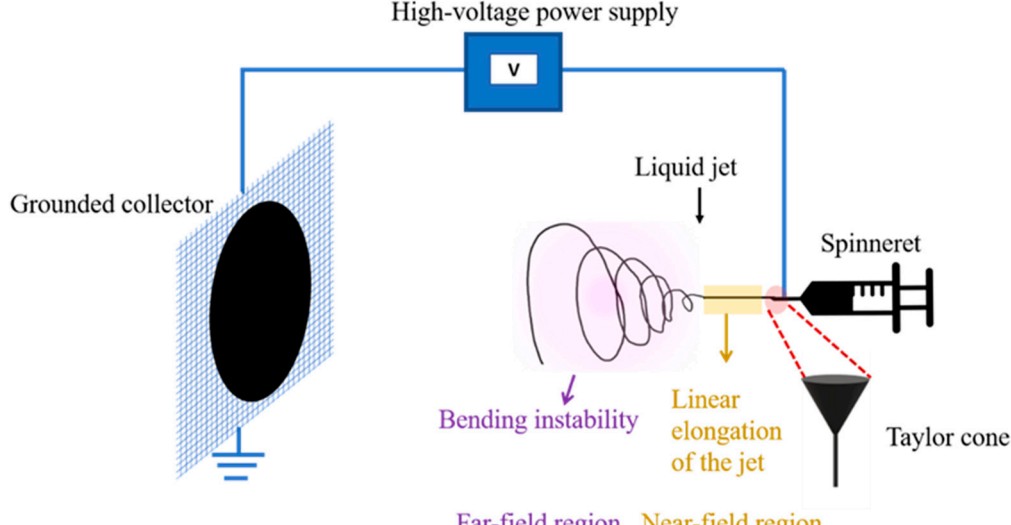

**Figure 6.** A scheme of the basic electrospinning setup [32].

### 2.3.3. Examples of Different Notions for Electrospinning

For a polymer melt or a single component polymer solution system, the ability to overcome the Rayleigh instability has been correlated to chain entanglements using the "solution entanglement number" by Shenoy et al., which is a semi-empirical methodology with an important assumption that chain entanglements are solely responsible for both the increase of the solution viscosity and the cause of the elastic network under the influence of an elongational strain [65]. The solution entanglement number, $(n_e)_{soln}$, is defined as the ratio of the weight-average molecular weight ($M_w$) to the entanglement molecular weight in solution ($(M_e)_{soln}$) [65]. The corresponding mathematical expression is given in Equation (2):

$$(n_e)_{soln} = \frac{M_w}{(M_e)_{soln}} = \frac{\phi M_w}{M_e} \tag{2}$$

$M_e$ is the entanglement molecular weight in the melt and $\phi$ is the polymer volume fraction. Generally, $M_e$ is a function of chain topology or geometry and $\phi$ relates to the dilution effect due to the presence of a solvent. Shenoy et al. [65] demonstrated that stable fiber formation for a neutral polymer in a good solvent occurs when $(n_e)_{soln}$ = 3.5 (or the number of entanglements per chain is greater than 2.5) depending on the polymer molecular weight, concentration, and solvent quality. This suggests that the polymer with high molecular weight has a large number of entanglements per chain, which results in better electrospinning behaviour. Many systems have displayed similar threshold of the number of entanglements per chain (>2.5) [66–69]. However, because of the underlying assumption of this approach, it is valid only for the single polymer system in a good solvent where polymer–polymer interactions are negligible [70]. Under a similar assumption, four solution regimes have been predicted for linear neutral polymers in a good solvent using a slope transition when plotting the log of specific viscosity ($\eta_{sp}$) as a function of the logarithm of the polymer concentration (c), including the dilute regime ($\eta_{sp} \sim c^{1.0}$), the semi-dilute unentangled regime ($\eta_{sp} \sim c^{1.25}$), the semi-dilute entangled regime ($\eta_{sp} \sim c^{4.8}$), and the concentrated regime ($\eta_{sp} \sim c^{3.6}$) [66]. The specific viscosity is expressed by Equation (3):

$$\eta_{sp} = (\eta_{soln} - \eta_{solv}) / \eta_{solv} \tag{3}$$

where $\eta_{soln}$ is the viscosity of the polymer solution and $\eta_{solv}$ is the viscosity of the solvent. The slope for each regime is also called the scaling value, which represents the contribution of the polymer concentration to the viscosity in a specific regime. The entanglement concentration ($c_e$) is defined as the polymer concentration at the transition from the semi-dilute unentangled to the semi-dilute entangled regimes [71,72]. The entanglement concentration has been used to study the electrospinning ability of the polymer solution and the corresponding morphology of the electrospun nanofibers [67]. It is found that beaded nanofibers formed at c = $c_e$, while defects and droplets disappear at c = 2 to 2.5 $c_e$. [67] Rubinstein [73] and Dobrynin [74] expands the theory to the salt-free linear polyelectrolyte solution, which predicts that the semi-dilute unentangled regime ($\eta_{sp} \sim c^{0.5}$), the semi-dilute entangled regime ($\eta_{sp} \sim c^{1.5}$), and the concentrated regimes ($\eta_{sp} \sim c^{3.75}$). It was found that a uniform electrospun fiber can be prepared from the salt-free polyelectrolyte solution at c = 8 $c_e$ − 10 $c_e$ [75].

The ability to maintain the jet formation during electrospinning can be provided by physical molecular interactions between polymer units or other molecules in the spinning solutions as polymer entanglements [65,76]. These physical molecular interactions include hydrogen bonding, electrostatic interactions, and hydrophobic effects [77]. Fiber formation that utilizes molecular interactions that can occur without high molecular weight polymers, under aqueous conditions, or at lower concentrations lead to advanced formulations for electrospinning. According to Ewaldz et al. [77] molecular interaction-driven electrospinning can be divided into three categories: polymer-polymer interactions, polymer–small molecule interactions, and supramolecular interactions shown in Figure 7. The first two categories are assigned to the system with one predominant molecular interac-

tion, whereas the last one, supramolecular interactions, is assigned to the system where constituents establish assemblies through multiple molecular interactions.

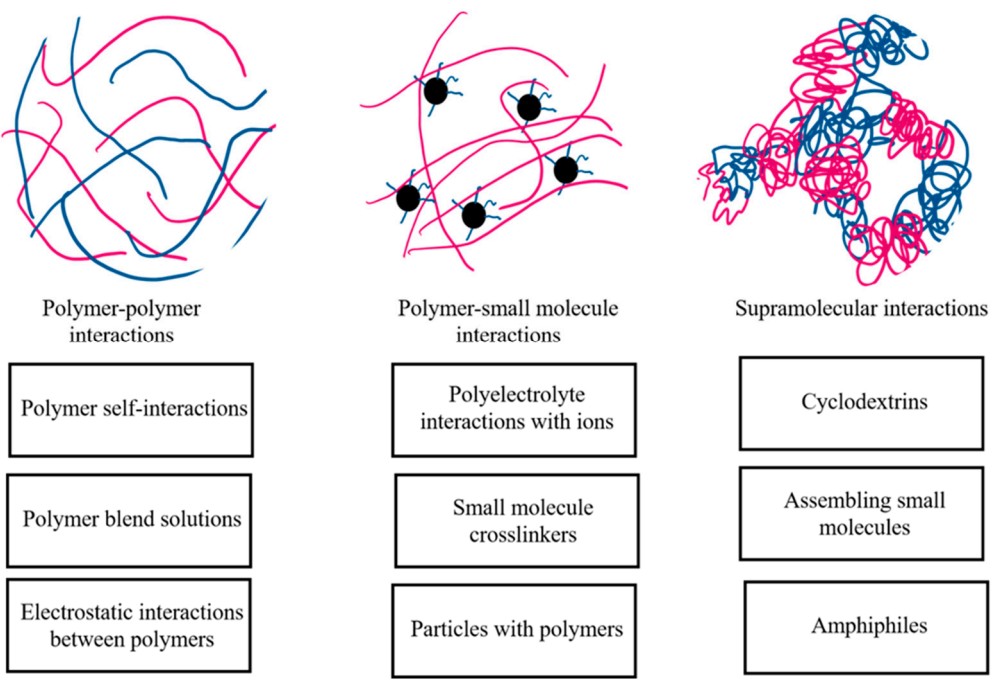

**Figure 7.** Categories of the molecular interaction-driven electrospinning. Reproduced with permission from Ewaldz et al. [77] Copyright (2019) American Chemical Society.

### 2.3.4. Polymer–Polymer Interactions

Among the various polymer–polymer interactions shown in Figure 7, most chitosan-based fiber composite materials prepared by electrospinning fall into the category of polymer blends. As discussed in Section 1.1, chitosan is a polyelectrolyte with different rheological properties and electrospinning behaviors that are compared with neutral polymers. The stiffness of chitosan polymer chains in the solution showed a weaker relationship between viscosity and concentration than neutral polymers in dilute and semi-dilute regimes. [74,75,78] However, because of the repulsion between charges, electrospinning of charged polymers requires higher concentrations, with $(n_e)_{soln} > 8$ compared to >2.5 for neutral polymers [75]. Furthermore, depending on the DA of chitosan, it may undergo gelation at a high concentration [79], which prevents chitosan from electrospinning at higher concentration values. Therefore, chitosan was blended with other high molecular weight polymers, like PEO, PVA, collagen, and pullulan, to achieve stable jet formation during electrospinning and to facilitate the formation of chitosan-based electrospun nanofibrous materials. For example, addition of high molecular weight (HMw; greater than 400 kDa) PEO to the chitosan solution is a promising strategy to promote the formation of chitosan electrospun nanofibrous materials [16–20,80,81]. Moreover, HMw PEO can act as a plasticizer for chitosan as flexible PEO chains which can occupy the void space among the rigid chitosan chains to reduce the inter- and intra- molecular interactions of chitosan chains that result in the prevention of the gelation of chitosan and reduction of the viscosity of the blends [80,81]. Collagen, pullulan, and PVA can also be used as additive polymers for blending with chitosan [22,33,82–84]. Blending of chitosan with collagen, pullulan, or PVA favours hydrogen bonding between the polymer subunits which leads to increased molecular interactions and stable jet formation during electrospinning [22,33,83]. Other hypotheses have been proposed for chitosan/collagen blends to explain for the improved electrospinning performance, which includes the enhancement of molecular interactions through chitosan assembly with collagen in a triple helix or the two polymers undergo complex formation via ionic interactions [77].

Chitosan-based electrospun fiber composite materials prepared using electrostatic interactions are rarely found in the literature. Electrospun fibers based on chitosan/hyaluronic acid coacervates represent a good example in this sub-category (electrostatic interactions between polymers), as shown in Figure 8 [85]. A stable suspension containing PECs between chitosan and hyaluronic acid was formed at 300–600 mM NaCl to control the rheology of the suspension. An alcohol mixture (methanol and ethanol; 5 wt.%) was used to promote solvent evaporation during electrospinning. As displayed in Figure 8, the viscosity of the PECs suspension decreased as the concentration of NaCl increased because of the salt-screening effect, resulted in attenuation of the electrostatic interactions between polymers. Moreover, salt concentration was found to affect the viscoelasticity of the PECs suspension, where the results indicate that PECs-based electrospun fiber systems can be manipulated by controlling the salt concentration.

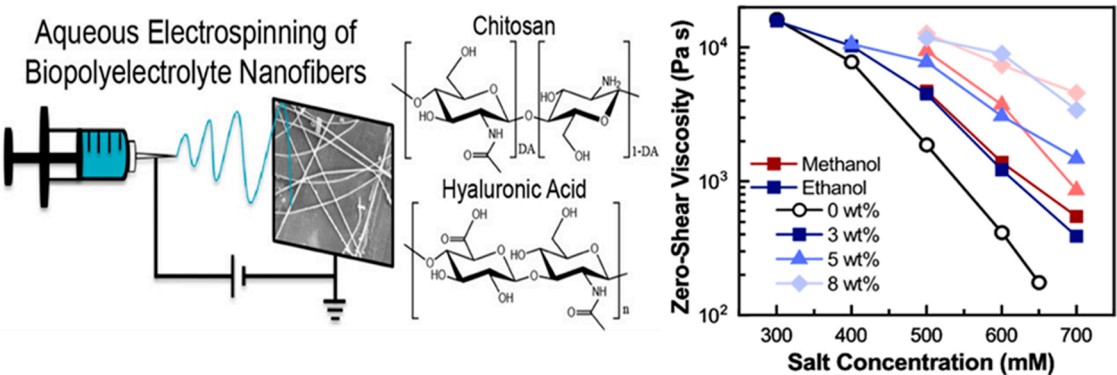

**Figure 8.** A plot of the zero-shear viscosity of chitosan/hyaluronic acid coacervates as a function of salt and methanol or ethanol concentration. Reproduced with permission from Sun et al. [85]. Copyright (2019) American Chemical Society.

Another example in this sub-category (electrostatic interactions between polymers) is chitosan/PAA electrospun fiber, as shown in Figure 9 [86]. The viscosity dependence on the weight fraction of chitosan is presented in Figure 9, where the log–log plot illustrates two different slope profiles that were observed at different weight fractions. The different slopes' suggest that chitosan-PAA interactions contribute to the viscosity at a low weight fraction of chitosan, whereas chitosan-chitosan interactions affect the solution's viscosity at a high weight fraction of chitosan. The optimal weight fraction between chitosan and PAA was found at 1:4.

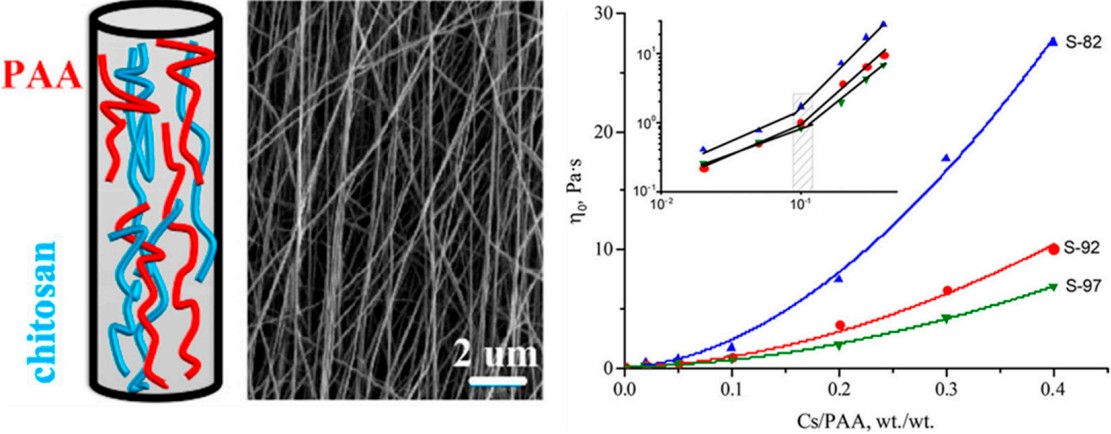

**Figure 9.** Viscosity vs. relative weight fraction of chitosan in different electrospinning solutions. The inset shows the same data redrawn on a log–log scale. Reproduced with permission from Zhang et al. [86]. Copyright (2018) American Chemical Society.

### 2.3.5. Supramolecular Interactions in Cyclodextrin-Based Systems

Single component hydroxypropyl-β-CD (HP-β-CD) [87] was reported to form electrospun fibers despite its relatively low molecular weight of HP-β-CD. The effect of fiber formation was attributed to the tendency of this systems to form supramolecular assemblies via hydrogen bonding. The role of hydrogen bonding in the system was studied by the addition of urea, as urea is known to disturb hydrogen bonding [87]. As a result, the viscosity of the solution decreased significantly, and the solution was no longer spinnable [87]. A rheological study was conducted that focused on HP-β-CD in dimethylformamide (DMF) by Zhang et al. [88] Based on the dependence of the specific viscosity on the concentration of the HP-β-CD, a transition from the unentangled ($\eta_{sp} \sim c^{11.0}$) and entangled regimes ($\eta_{sp} \sim c^{15.5}$) was shown. In turn, the scaling values for both regimes were higher than those for neutral polymers in a good solvent, ($\eta_{sp} \sim c^{1.25}$) and ($\eta_{sp} \sim c^{4.8}$), respectively. However, the scaling value for HP-β-CD entangled regime was in good agreement with the concentrated regime ($\eta_{sp} \sim c^{14.0}$) of the surfactants [88]. This suggests that the HP-β-CD solution has similar colloidal behavior to worm-like micelles, which is considered to contribute to its electrospinning properties [88]. An alternative mechanism for electrospinning of HP-β-CD in pure aqueous solutions was proposed by Manasco et al. [89]. Based on their study [89], the electrospinning jet was stabilized by the water-HP-β-CD assemblies formed through the strong hydrogen bonding. The effect is supported by the fact that there was almost 0% free water in 60% (*w/w*) HP-β-CD solution, a favorable concentration was observed for electrospinning. This study also demonstrated that HP-β-CD solutions in water had high viscosity but not high elasticity. This is in contrast to the previous report that high elasticity is a condition for electrospinning solutions without polymer entanglements [68]. A recent study of chitosan/trifluoroacetic acid (TFA)/HP-β-CD system was conducted by Xue and Wilson [90] where several components were found to form a multi-component assembly (cf. Figure 10) attributed to facilitating this system's propensity for electrospinning. TFA serves as a joint between chitosan and HP-β-CD in this assembly: TFA electrostatically bonds with chitosan ($-COO^- —-NH_3^+$) and forms a host-guest "facial" complex between the trifluoromethyl groups of TFA with HP-β-CD ($-CF_3$—- HP-β-CD) [90]. The unique supramolecular assembly between the HP-β-CD units with the TFA anion was supported by results from previous studies of host-guest complexes between β-CD and haloalkanes such as halothane by Wilson and Verrall [91], further highlighting the role of supramolecular interactions in such complex multi-component systems.

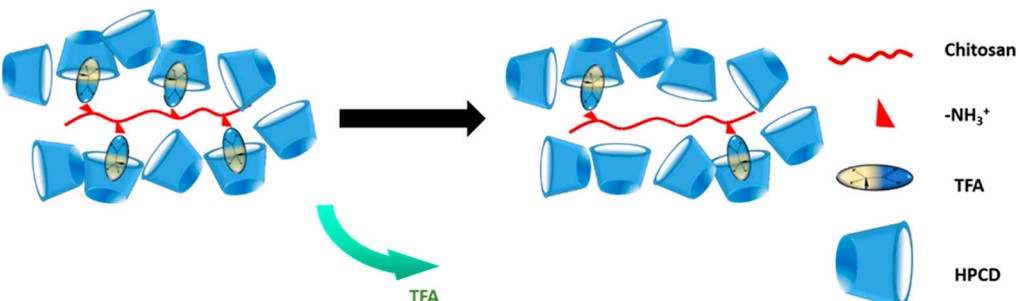

**Figure 10.** An illustrative view of the compositional change of a chi:HP-β-CD electrospun fiber over time, where the green arrow shows incremental temporal loss of trifluoroacetic acid (TFA). Reprinted with permission from [90].

Overall, it is worth noting that, because of the underlying assumption of the chain entanglement, the entanglement concentration, and the scaling value acquired from the polymer system of interest cannot be used to foresee the electrospinning ability of the system and the morphology of the corresponding electrospun nanofiber for many systems, as mentioned above [88,92–97]. Nevertheless, the entanglement concentration and the scaling value, especially the scaling value compared with the theoretical value of the linear

neutral or polyelectrolyte polymer system [88,92–97] provides valuable information on the microstructure of polymer solutions of interest that undergo electrospinning.

### 2.4. Post-Treatment Methods

### 2.4.1. Purpose of Post-Treatment

Once chitosan-based composite fibrous materials are obtained, post-treatment is sometimes required to increase the stability of the materials in an aqueous medium. There are two categories for post-treatment of chitosan fibrous materials: chemical and physical methods [4]. For chemical methods, crosslinking through a covalent bond is typically used to reduce the number of free hydrophilic groups in the chitosan (amino group or hydroxy group), increasing the stability of chitosan in water. Some commonly used crosslinking agents are epichlorohydrin [98] or glutaraldehyde [99]. Although a study conducted by Zhao et al. [100] has shown that the chemical crosslinking does not change the biocompatibility of the chitosan, however, the safety of chemically cross-linked chitosan is still of concern due to the presence of the unreacted or partially reacted cytotoxic crosslinking agent in the product. For physical methods, the chitosan-based material is crosslinked via electrostatic interactions by multivalent anions, like oxalic acid [101], citric acid [102–104], or tripolyphosphate [105,106]. Another physical method often used is thermal treatment [84,107–109]. Physical methods are favorable for preparing materials for applications such as wound dressing, tissue engineering, and food packaging because there are no questionable or harmful organic chemicals used in the process.

### 2.4.2. Emerging Approaches

### Cinnamaldehyde as a Cross-Linking Agent

Cinnamaldehyde has been used to crosslink a wheat protein [110], which is known as a low toxic crosslinking agent [84,110]. Recently, cinnamaldehyde was applied to cross-link chitosan/pullulan (CCP) electrospun fiber to enhance the hydrophobicity of the composite fiber leading to improved water stability and better vapor barrier properties, as shown in Figure 11 [84].

### Coagulant Bath Combined with Physical Cross-Linking via Phosphate Ions

According to Dodero et al., [111] chitosan/PEO electrospun composite fiber was firstly treated in the coagulant bath containing ethanol/$NH_2OH$/$H_2O$ with a volume ratio of 7/2/1 (pH = 7.5) to deprotonate amine groups of chitosan and remove PEO (the co-electrospinning agent). The treated composite fiber was then cross-linked by $Na_2HPO_4$ and ethylene glycol diglycidyl ether (EGDE), respectively. The composite fiber's water stability was ionically cross-linked by phosphate ions which was compared to the one chemically cross-linked by EGDE, as demonstrated in Figure 12. The results suggest that the ionically cross-linked composite fiber has similar water stability as the chemically cross-linked product. Therefore, the composite fiber treated with this method will favor biomedical applications.

### Thermal Treatment Combined with Chemical Cross-Linking

In the study conducted by Mak and Leung [112], chitosan/PEO electrospun composite fiber was treated using the thermal treatment and chemical cross-linking to at-tain advantages from both post-treatment methods (cf. Table 1 in Ref. [112]). The pro-posed chemical structure involved in this study was illustrated therein (cf. Figure 2 in Ref. [112]). The outcome of applying thermal and chemical treatment is that genipin cross-linked chitosan/PEO electrospun fibers after thermal treatment (CSGA) displays reduced free amine groups, increased hydrophobicity, improved water and thermal stability, enhanced tensile strength, and stability against lysozyme. However, CSGA suffers from attenuated antibacterial activity due to the reduction of the free amine group accessibility.

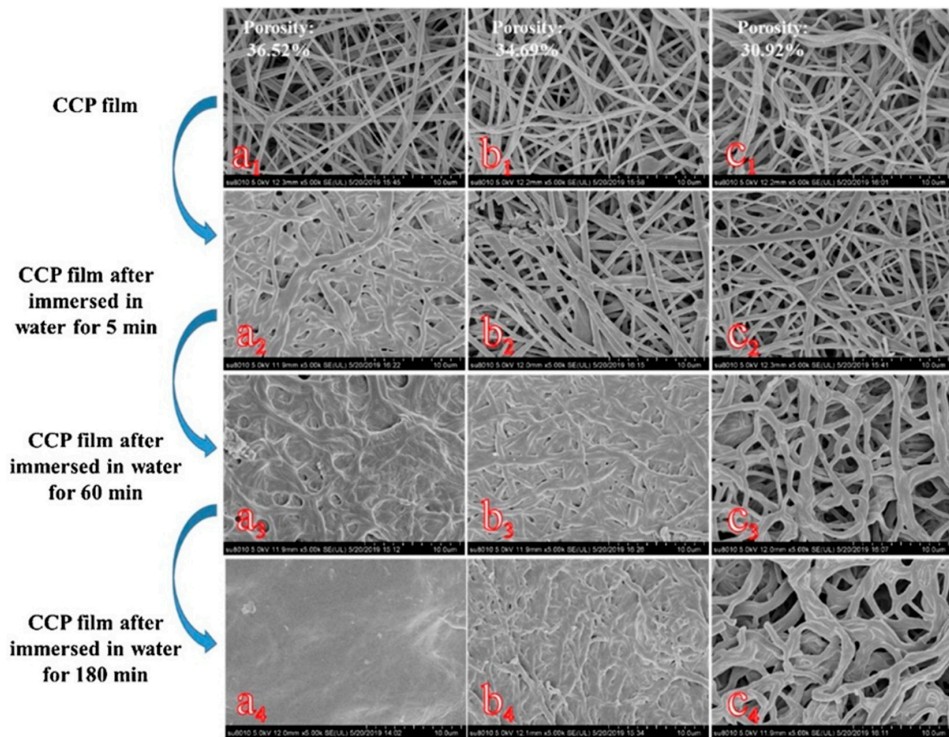

**Figure 11.** SEM images of cinnamaldehyde cross-linked nanofiber films with chitosan/pullulan weight ratios of 1/9 (CCP19), 2/8 (CCP28) and 3/7 (CCP37) ((**a₁**), CCP19; (**b₁**), CCP28; (**c₁**), CCP37) and SEM images of different CCP films after immersing in water for 5 min ((**a₂**), CCP19; (**b₂**), CCP28; (**c₂**), CCP37), 60 min ((**a₃**), CCP19; (**b₃**), CCP28; (**c₃**), CCP37) and 180 min ((**a₄**), CCP19; (**b₄**), CCP28; (**c₄**), CCP37). Reprinted with permission from [84].

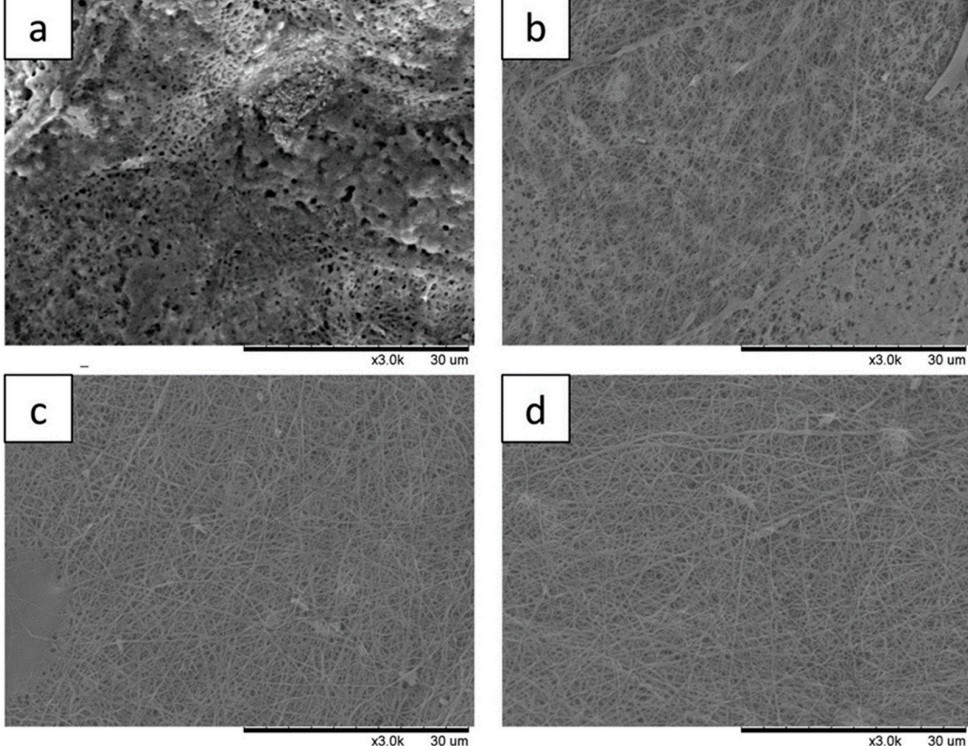

**Figure 12.** SEM images of chitosan-based mats crosslinked with (**a**) 5% *w/v* Na$_2$HPO$_4$ solution, (**b**) 10% *w/v* Na$_2$HPO$_4$ solution, (**c**) 2.5% *v/v* EGDE solution, and (**d**) 5% *v/v* EGDE solution. Reprinted with permission from [111].

**Table 1.** Summary of emerging approaches and examples for chitosan-based fiber composite materials.

| Emerging Examples | Innovation | Applications |
|---|---|---|
| **Examples of freeze-drying** | | |
| cm-cellulose/HACC [50] [1] | Salt concentration impacts morphologies of fibers. | Wound dressing |
| Chitosan/gelatin [51] | Thermal treatment applied before freeze-drying. | Tissue scaffolds |
| **Examples of electrospinning** | | |
| Chitosan/pullulan [84] | Pullulan acts as a green co-electrospinning agent. | Active packaging |
| Chitosan/alginate [85] Chitosan/PAA [86] [2] | PECs suspension is used for electrospinning | Biomedical applications pH-responsive coating for nanodevices |
| **Post-treatment methods** | | |
| Cinnamaldehyde [84] | A low toxic chemical crosslinking agent | Active packaging |
| Coagulant bath combined with physical crosslinking [111] | Use of phosphate as a physical cross-linking agent and eliminating the co-spinning agent | Wound healing patches/ drug delivery systems |
| Thermal treatment combined with chemical crosslinking [112] | Synergized advantages from both methods | Wound healing/ tissue scaffolds |

[1] cm-cellulose/HACC: carboxymethyl cellulose/N-2-hydroxylpropyl trimethyl ammonium chloride chitosan. [2] PAA: poly(acrylic acid).

## 3. Future Perspectives and Concluding Remarks

### 3.1. Recent Developments on the Preparation Procedures of Chitosan Composites

Emerging examples and approaches for chitosan-based fiber composite materials are summarized in Table 1. Recent advances in the mode of preparation for such composite materials has focused mainly on improving the biocompatibility, stability in aqueous media, controlling the morphology, and enhancing the mechanical properties. In the case of freeze-dried materials, the main target is controlling the morphology and enhancing the mechanical properties, which can be accomplished by changing reaction conditions such as salt concentration [50] and heating before freeze-drying [51]. For electrospun materials, an emphasis on improving the biocompatibility and water stability by removing the co-spinning agent [111] or using a biocompatible co-spinning agent to replace common synthetic polymers [84]. Suspensions of chitosan-based PECs represent a new group of starting materials for chitosan electrospun composite materials. Such types of electrospun materials can be prepared in aqueous media that possess various pH-dependent properties [85,86]. Moreover, post-treatment methods that combine physical methods and low or non-toxic cross-linkers have been reported [84,111,112].

### 3.2. Future Perspectives

Key criteria for the selection of new chitosan-based fiber composite materials require biocompatibility, sustainability, and good water stability among other structural and physicochemical properties. Suitable post-treatment methods, such as chemical cross-linking of materials with biocompatible agents and physical cross-linking is required to achieve the desired stability outcomes for electrospun fiber systems. After guaranteeing the final product's biocompatibility and water stability, the future design of such fiber composite will emphasize the precise control of the morphology of the final product's and improvement of the mechanical properties. One approach for improving the mechanical properties of chitosan composite fibers involves the incorporation of carbon nanotubes (CNTs) or carbon fibers into the structure of the chitosan-based composite [113–118]. One critical procedure for preparing chitosan composite fibers that contain CNTs is to formulate a homogenous dispersion for electrospinning, which determines the resulting products' morphology, mechanical, and electric properties [114,115,117,119,120]. In particular, the use of CNTs may serve as a macromolecular structural template [114] that results in composite fiber materials with reduced diameter and greater uniformity of the resulting composite fibers (cf. Figure 7 in [114]). Supramolecular-based templation effects of this type were reported [121] for self-assembled chitosan-polyaniline (PANI) binary composites [121]. The nature of these

chitosan-PANI composites were subsequently used as electrochemically active catalyst supports for stabilizing Ag nanoparticles for the enhanced reduction of 4-nitrophenol under mild conditions [122]. The utility of supramolecular-based approaches described herein (cf. Section 2.3.3) represent a fertile research area with considerable scope for future development, which builds upon the various bottom-up synthetic strategies described herein [77,90]. The rational design of biopolymer fiber composites with controlled architectures is likely to catalyze further studies aimed at establishing improved *structure–property* relationships. In turn, the ability to tailor the hydration properties through understanding the hydrophile-lipophile character of biopolymer composite fibers is likely to have a significant impact in various biomedical fields, such as advanced drug delivery, bone tissue engineering, and biosensor technology [123–132]. Continued interest in the development of biopolymer fiber composites will serve to address the UN Sustainable Development Goals (SDGs) [133] since the development of suitable alternatives to conventional plastics and non-renewables is a key SDG goal. Concern over the buildup of single use plastics in the environment and inadequate waste disposal strategies can be addressed through the development of sustainable alternatives such as biopolymer composites. Significant ongoing interest on the development of biopolymer composites for water and wastewater treatment applications represent key areas of ongoing active research. Biopolymer fiber systems have proven to be an effective adsorbent morphology, as compared with other types of conventional powders and bead-based systems for environmental remediation [125]. In a recent study, the role of acetic acid on the structure and physicochemical properties of chitosan films was reported [134], where the dissolution of natural chitosan films resulted in marked changes to the biopolymer structure. Evidence of such effects was established by complementary materials characterization and quantum chemical calculations, where variations in the surface chemistry and morphology of the chitosan films occurred upon acid treatment (cf. Figures 2–4 in [134]). This new insight highlights the importance of the source of chitosan and the influence of pre-treatment prior to electrospinning applications. In turn, various types of chemical treatment are likely to affect the native fibril structure, morphology, and physiochemical properties of fiber composites, as evidenced by examples reported herein. Hence, future research directed at natural chitosan films and fiber mats is likely to have a significant impact on diverse applications in the fields of pharmaceutics, cosmetics, agriculture, environmental science, and biomedicine. Therefore, further research is needed to understand the formation mechanism of such chitosan homogenous dispersions, which is forecasted to lead to the development of robust protocols to yield reproducible electrospun products. In turn, such products will allow for an improved understanding of the *structure–function* properties of biopolymer composite systems [32,77,90,134].

### 3.3. Conclusions

This review focuses on different preparation procedures and methods that involve freeze-drying and electrospinning of chitosan composite fibrous materials and the consequence of such protocols on the material structure. Examples of the importance of supramolecular chemistry in electrospinning is evidenced in biopolymer–biopolymer, biopolymer–small molecule, and molecular system that undergo self-assembly such as amphiphiles, cyclodextrins, and other small molecules [77,90]. A coverage of various examples that highlight the preparation of chitosan-based composite materials through the use of chitosan and its modified forms, along with post-treatment of the fiber composites were described. The underlying rationale regarding the various materials design was presented to provide a molecular-level perspective. The design of chitosan-based fibers included an overview of the background on freeze-drying and electrospinning methods. Examples associated with these methods were discussed to highlight the key aspects of the experimental conditions to achieve such fiber systems. A greater understanding of the different preparation methods is anticipated to contribute to composite fiber materials that possess well-defined molecular architectures. In turn, an improved understanding of the *structure–function* relationships of biopolymer fiber composites is anticipated, which can

afford greater insight on various physicochemical properties such as hydration [135–137]. Fundamental studies of this type concerning the hydration of chitosan composite fibrous materials obtained via a controlled design will yield new materials that can be tailored to suit unique fiber-based technological applications, which range from biomedicine to environmental science [32,123–131].

**Author Contributions:** Conceptualization, C.X.; investigation, C.X.; resources, L.D.W.; writing—original draft preparation, C.X.; writing—review and editing, C.X. and L.D.W.; visualization, C.X.; supervision, L.D.W.; funding acquisition, L.D.W. All authors have read and agreed to the published version of the manuscript.

**Funding:** This research was funded by the Natural Sciences and Engineering Research Council (Discovery Grant Number: RGPIN 2016-06197) and the University of Saskatchewan.

**Institutional Review Board Statement:** Not applicable.

**Informed Consent Statement:** Not applicable.

**Data Availability Statement:** The data presented in this study are available within the article.

**Acknowledgments:** C.X. acknowledges the University of Saskatchewan for supporting aspects of this research through contributions from the scholarship and Teaching Assistantship program in the Department of Chemistry.

**Conflicts of Interest:** The authors declare no conflict of interest.

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
