# Peer review of "An Overview of the Design of Chitosan-Based Fiber Composite Materials"

_jcs, doi:10.3390/jcs5060160_

Round 1

Reviewer 1 Report

The review article is written very well. The aim of the paper has been defined and presented within the framework of this paper. Particularly noteworthy is the literature introduction, which has been described at a great depth.  The results of the research and the description of the research methodology also do not raise any doubts. The work is suitable for publication, after taking into account minor remarks:

  1. Since the introduction contains information on the subject of the application of carbon fibres materials, I recommend considering valuable papers - such as: 10.1016/j.tws.2020.106627, 10.1016/j.compstruct.2015.06.058.
  2. Please clearly demonstrate the novelty of this paper relative to most similar research papers, and describe your new contribution to the research.
  3. The conclusions are far too short. Conclusions should clearly present a qualitative and quantitative assessment of the research conducted by the investigators within the review article.

Author Response

Authors’ Response to Reviewer Reports on MD ID:  jcs-1261182

Reviewer #1:

The review article is written very well. The aim of the paper has been defined and presented within the framework of this paper. Particularly noteworthy is the literature introduction, which has been described at a great depth.  The results of the research and the description of the research methodology also do not raise any doubts. The work is suitable for publication, after taking into account minor remarks:

  1. Since the introduction contains information on the subject of the application of carbon fibres materials, I recommend considering valuable papers - such as: 10.1016/j.tws.2020.106627, 10.1016/j.compstruct.2015.06.058.

Response: The authors appreciate the reviewer’s comment. We have included refererence to these papers and additional citations (>20) are included in the remarks and conclusion section.

  1. Please clearly demonstrate the novelty of this paper relative to most similar research papers, and describe your new contribution to the research.

Response: In this contribution several reviews were cited (cf. Refs. 5, 14, 15, 21, 53, 77). The novelty of this review relates to various modes of preparation of chitosan fiber composites that include freeze-dried and electrospun composites. This review provides insight on the underlying rationale relating to tuning of the physicochemical properties related to the hydrophile-lipophile balance of such composites, and control over the structure and morphology of the resulting materials. In comparison with the reviews cited herein, much of the coverage relates to the general aspects of the fiber attributes and their practical applications.

  1. The conclusions are far too short. Conclusions should clearly present a qualitative and quantitative assessment of the research conducted by the investigators within the review article.

Response: The authors appreciate the reviewer’s comment. We modified the conclusion section accordingly to address this concern.

In summary, the authors’ appreciate the insightful and constructive comments provided by Reviewer #1. We have carried out comprehensive editing of the manuscript to address language, syntax, and clarity throughout in order to meet the high standards of the Journal of Composites Science.

Reviewer 2 Report

This review contributes to the understanding of chitosan-based composite fibrous materials prepared by freeze-drying and electrospinning.

This article is well written. Following is some suggestions for improving the article:

Check L30-32: How can the chitin deacetylases obtain chitosan oligomers?

L89-95: What are the reaction conditions for the heterogeneous method? What’s the DA for the homogeneous method?

L98-117: Provide a figure to illustrate different forms of chitosan.

L122-123: Why poor moisture barrier properties of chitosan are important for food packaging? In fact, strong moisture barrier properties are important for food packaging.

L129: What do you mean “cm-chitosan”?

Author Response

Authors’ Response to Reviewer Reports on MD ID:  jcs-1261182

Reviewer #2:

This review contributes to the understanding of chitosan-based composite fibrous materials prepared by freeze-drying and electrospinning.

This article is well written. Following is some suggestions for improving the article:

Check L30-32: How can the chitin deacetylases obtain chitosan oligomers?

Response: The authors appreciate the reviewer’s comment. The process that produces chitosan oligomers by chitin deacetylases was reviewed by Tsigos et al. (Trends Biotechnol. 2000, 18, 305–312)

Tsigos, I.; Martinou, A.; Kafetzopoulos, D.; Bouriotis, V. Chitin deacetylases: New, versatile tools in biotechnology. Trends Biotechnol. 2000, 18, 305–312.

L89-95: What are the reaction conditions for the heterogeneous method? What’s the DA for the homogeneous method?

Response: The manuscript was revised accordingly.

L98-117: Provide a figure to illustrate different forms of chitosan.

Response: The use of different forms of chitosan were not the focus this review. In general, we have provided coverage of articles that utilize commercially available chitosan that possess typical average degree of acetylation. It is well established that different morphological forms of chitin/chitosan can be obtained from variable biomass sources (e.g., insects, fungi, lichens, crustaceans). A reference to the use of chitosan from different sources (see Ref. [135] https://doi.org/10.1021/acssuschemeng.0c06373) is noted in the future perspectives section to address the reviewer’s valuable suggestion.

L122-123: Why poor moisture barrier properties of chitosan are important for food packaging? In fact, strong moisture barrier properties are important for food packaging.

Response: The authors appreciate the reviewer’s comment. The original wording was misleading. We have made revisions accordingly to address the reviewer query.

L129: What do you mean “cm-chitosan”?

Response: cm-chitosan is an abbreviation for carboxymethylated chitosan, as defined in section 2.1.

In summary, the authors’ appreciate the insightful and constructive comments provided by Reviewer #2. We have also carried out comprehensive editing of the manuscript to address language, syntax, and clarity throughout in order to meet the high standards of the Journal of Composites Science.
